# On the Variability of Functional Connectivity and Network Measures in Source-Reconstructed EEG Time-Series

**DOI:** 10.3390/e23010005

**Published:** 2020-12-22

**Authors:** Matteo Fraschini, Simone Maurizio La Cava, Luca Didaci, Luigi Barberini

**Affiliations:** 1Department of Electrical and Electronic Engineering, University of Cagliari, 09123 Cagliari, Italy; s.lacava@studenti.unica.it (S.M.L.C.); ldidaci@unica.it (L.D.); 2Department of Medical Sciences and Public Health, University of Cagliari, 09123 Cagliari, Italy; barberini@unica.it

**Keywords:** EEG, source analysis, functional connectivity, network

## Abstract

The idea of estimating the statistical interdependence among (interacting) brain regions has motivated numerous researchers to investigate how the resulting connectivity patterns and networks may organize themselves under any conceivable scenario. Even though this idea has developed beyond its initial stages, its practical application is still far away from being widespread. One concurrent cause may be related to the proliferation of different approaches that aim to catch the underlying statistical interdependence among the (interacting) units. This issue has probably contributed to hindering comparisons among different studies. Not only do all these approaches go under the same name (functional connectivity), but they have often been tested and validated using different methods, therefore, making it difficult to understand to what extent they are similar or not. In this study, we aim to compare a set of different approaches commonly used to estimate the functional connectivity on a public EEG dataset representing a possible realistic scenario. As expected, our results show that source-level EEG connectivity estimates and the derived network measures, even though pointing to the same direction, may display substantial dependency on the (often arbitrary) choice of the selected connectivity metric and thresholding approach. In our opinion, the observed variability reflects the ambiguity and concern that should always be discussed when reporting findings based on any connectivity metric.

## 1. Introduction

The idea to estimate the statistical interdependence among (interacting) brain regions, generally named as functional connectivity [1,2,3], has motivated numerous researchers to investigate how the resulting networks may organize themselves, in the context of the importance of the whole [4], under any conceivable scenario. This phenomenon seems of particular relevance because brain function not only critically depends on functional segregation, but also on functional integration, which, indeed, relates to the pattern of interactions between brain regions [5]. In general, functional connectivity may be investigated both at scalp- and at source-level. Nevertheless, it has been extensively shown that the two different approaches may lead to important differences in the reported results [6,7], as at scalp-level the EEG signals are more corrupted by effects of field spread. Even though this problem cannot be considered completely absent at source-level, it seems to be importantly attenuated in this latter case [8].

Countless studies reported how these patterns of statistical interdependence, computed with several and different metrics, can be associated with behavioral/clinical parameters or used to contrast different groups/conditions [9]. Even though this idea is not at initial stages, its practical application is still far to be widespread. One concurrent cause may be related to the proliferation of different approaches (metrics) to estimate the statistical interdependence among these signals [5,10,11,12]. Despite their substantial differences, all these metrics aim to catch the underlying statistical interdependence among the (interacting) units. The tacit idea that all these metrics can be used interchangeably because they measure the same connectivity may induce to inaccurate interpretations. Here, we want to investigate whether the arbitrary choice of the connectivity metric may have a severe impact on the results in a realistic scenario. Indeed, it is well known that different metrics could measure different characteristics, elements or aspects of the underlying connectivity, making it very difficult to define the ‘true’ connectivity. The issue of using the same name (i.e., functional connectivity) for all the different approaches has probably contributed to generate confusion and to hinder the comparison among different studies, since at the end they are based on different principles: linear or nonlinear relations, time or frequency domain, amplitude or phase information. Moreover, they have also been tested and validated using different methods, simulation [13,14] or empirical studies [12,13], making it even more difficult to understand to what extent they are similar or not.

In this study, we aimed to compare a set of different metrics commonly used to estimate the functional connectivity on a public EEG dataset [15,16] representing a possible realistic scenario. Ten different connectivity metrics were included in the analysis, together with five different thresholding approaches used in order to investigate several commonly used network measures. The proposed scenario consists of contrasting two different resting-state EEG conditions, namely eyes-closed and eyes-open, on 109 subjects recorded with a 64-channel system. The EEG signals were successively reconstructed at source-level and projected onto the Desikan–Killiany atlas [17]. Other than the inherent computational differences among the connectivity metrics, it is relevant to highlight that other methodological issues may have an effect on the reported findings, as for example, the problem of field spread, volume conduction, and reference montages [18]. For this reason, we decided to perform the analysis using metrics that are more prone to an erroneous estimate of connectivity and metrics that tend to limit these effects prior to computing the connectivity, including phase-based metrics that are less sensitive to these spurious interactions [18]. Moreover, since network density (the number of connections in a network) will directly influence the estimated network measures [19], we performed the analysis using four different densities (preserving 10%, 15%, 50% and 100% of the weights) and two methods of filtering information in the complex brain network that helped to overcome the problem of network density in the network analytical studies, namely the minimum spanning tree (MST) [20] and the efficiency cost optimization (ECO) [21]. In line with the investigated scenario, where we contrast eyes-closed and eyes-open conditions, all the reported results refer to the alpha [8–13 Hz] frequency band that should be a considerable marker of the underlying differences. All the analysis was performed using MNE python software [22] and Brain Connectivity Toolbox for MATLAB [23].

## 2. Material and Methods

### 2.1. Dataset

In order to test our hypothesis, we used a public and freely available EEG dataset [15,16] consigning on a set of recordings performed on 109 subjects, including signals from resting-state for eyes-closed and eyes-open recordings, each one lasting 1 min. The EEG traces were recorded from 64 electrodes as per the international 10-10 system with a sampling frequency equals to 160 Hz. All the EEG recordings are available at the following link: https://physionet.org/content/eegmmidb/1.0.0/.

### 2.2. Preprocessing

The EEGLAB toolbox (version 13_6_5b) [24] was used to re-reference to common average reference. Successively, ADJUST (version 1.1.1) [25], a fully automatic algorithm based on Independent Component Analysis (ICA), was used to detect and remove artifacts from the EEG signals. Subsequently, the source-based EEG signals were reconstructed using Brainstorm software (version 3.4) [26] with the head model created using a symmetric boundary element method in Open-MEEG (version 2.3.0.1) [27] based on the anatomy ICBM152 brain. The whitened and depth-weighted linear L2 minimum norm estimate (wMNE) [28] was used with an identity matrix as noise covariance. The source-reconstructed EEG time-series were projected onto the Desikan-Killiany atlas [17], which includes 68 regions of interest and where the time-series for voxels within a ROI were averaged after flipping the sign of sources with opposite directions. The subsequent analysis was performed using five non-overlapping epochs of 12 seconds, which is in line with what reported in [29].

### 2.3. Connectivity Metrics

Ten different connectivity metrics have been included in the analysis. In particular, for each subject and each condition we computed, for the alpha [8–13 Hz] frequency band, the following metrics: coherence (coh) [11], coherency (cohy) [30], imaginary coherence (imcoh) [30], phase-locking value (plv) [31], corrected imaginary PLV (icplv) [32], pairwise phase consistency (ppc) [33], phase lag index (pli) [34], unbiased estimator of squared PLI (pli2_unbiased) [35], weighted phase lag index (wpli) [35] and the debiased estimator of squared WPLI (wpli2_debiased) [35]. All the metrics were computed using the function mne.connectivity.spectral_connectivity from the MNE python software [22]. It is known that the unbiased procedure used to estimate both the pli2_unbiased and the wpli2_debiased may lead to negative values. In this study we have tested different solutions (i.e., round to zero the negative values or normalize to the 0–1 range) that have led to very similar results.

### 2.4. Network Measures

The network analysis was performed using four different densities, FWEI (100% of weights preserved), WEI10 (10% of weights preserved), WEI15 (15% of weights preserved), and WEI50 (50% of weights preserved). Despite these thresholding procedures are far to represent an optimal solution [19], they are still very commonly used in network community. Furthermore, two methods to filtering information in complex brain network intended to overcome the problem of network density in network analytical studies, namely the minimum spanning tree (MST) [20] and the efficiency cost optimization (ECO) approach [21] were also added to the analysis. This analysis was performed using the Brain Connectivity Toolbox for MATLAB [23].

### 2.5. Cluster Analysis

The cluster analysis was used to investigate the possible natural clusters, without any ‘a priori’ assumptions, with an unsupervised approach to reveal the possible existence of different functional connectivity groups. For each connectivity metric and each subject, a feature vector, containing the connectivity profile (extracted as the triangular connectivity matrix), was obtained. The clustering approach was based on a k-means method, using the k-means++ algorithm for centroid initialization and squared Euclidean distance. A similar analysis was recently conducted on a smaller set of connectivity metrics [36]. The silhouette analysis [37], which can be employed to study the separation distance between the clusters, was used to define the optimal number of clusters. In particular, this analysis allows us to understand how well each object lies within its cluster by comparing the similarity between an object and its own cluster (cohesion) versus the similarity between an object and other clusters (separation), where the higher the silhouette value the better the objects are well matched to their own cluster. In this study, we used the mean silhouette value over all points to measure how appropriately the data have been clustered. Successively, to evaluate the clustering quality on the basis of the discovered common properties between the different connectivity metrics, the purity evaluation measure was used. To compute purity, each cluster is assigned to the most frequent class in the cluster, and then the accuracy of this assignment is measured by calculating the number of correctly assigned objects divided by the numerosity of the cluster. Bad clustering purity value tends to 1/n where n indicates the number of classes, perfect clustering has a purity of 1. Finally, to investigate the possibility that the natural groupings may result in clusters which include more than one metric, we associated to each metric a pseudo-label that is the index of the cluster in which this measure is most represented. If $n_{i,\;j}\;$ is the number of connectivity profiles of measure *i* present in cluster *j* we can define $pseudo\_label_{i}=\arg\underset{j}{\max}\left(n_{i,j}\right)$. In other words, if most of the connectivity profiles of the metric *i* belongs to the cluster *j*, we associated the pseudo-label *j* to this metric. The purity computed with the new ‘pseudo-labels’ for several k values are later reported.

### 2.6. Statistical Analysis

In order to contrast the two conditions, namely eyes-closed and eyes-open resting state, separately for each connectivity metric, the Wilcoxon signed-rank test was used. The statistical results were reported in terms of *p*-value, effect size, and direction of the effect. The alpha level, equals to 0.05, was corrected for the number of measures extracted for each analysis.

## 3. Results

### 3.1. Global Connectivity Patterns

In order to have a reference for the different connectivity metrics, the global connectivity patterns (averaged over all the subjects) for each metric and for each condition (eyes-closed and eyes-open) are depicted in Figure 1. We decided to depict the connectivity patterns using different scales across the different metrics to simplify the visual inspection of those differences. The distribution of connectivity weights for each metric (averaged over the 109 subjects) are depicted in Figure 2.

### 3.2. Cluster Analysis

The cluster analysis, performed to investigate the existence of possible natural clusters, without any ‘a priori’ assumptions on possible grouping, was conducted separately for each experimental condition (i.e., eyes-closed and eyes-open resting-state). In Figure 3 we show the set of connectivity profiles (used as feature vectors) for each connectivity metric and each condition. The mean silhouette values for all possible k values ranging from 2 to 10 (i.e., the number of connectivity metrics) are summarized in Table 1. The mean silhouette value is a measure of how appropriately the data have been clustered. If each metric was represented in a different cluster, we would expect the higher silhouette value for k = 10. Conversely, for k = 10 we have observed the lowest silhouette value, whilst the higher value is obtained for k = 2. These findings represent a strong evidence that the clustering obtained with k = 10 doesn’t represent the correct, natural aggregation of the data. The purity values for k = 10 are reported in Table 2 and confirmed that while some clusters are predominantly populated by the elements of a single measure (see clusters 2, 4, 8, 10 for the eyes-closed condition and clusters 1, 2, 4, 9, 10 for the eyes-open condition), other clusters show a mixture of metrics. For example, in the clusters 1, 3, 5 for the eyes-closed condition the most represented metric constitutes less than 40% of the cluster elements.

These results encourage us to investigate the possibility that the natural groupings, if they exist, are less than 10, with each natural group composed of more than one measure. To verify this hypothesis’s correctness, as described in the Section 2.5, we associate to each measure a pseudo-label (i.e., the index of the cluster in which this measure is most represented) and the corresponding purity values, for different k, are summarized in Table 3.

### 3.3. Network Analysis

The results from the FWEI approach (where the 100% of weights were preserved) are summarized in Table 4. A significant difference between the two conditions in global efficiency was observed for all the connectivity metrics. Moreover, all the connectivity metrics allowed to observe a significant difference for the clustering coefficient and for the modularity. The results from the WEI10 approach (where the 10% of weights were preserved) are summarized in Table 5. A significant difference between the two conditions in global efficiency was observed for two out of the ten connectivity metrics, namely cohy (*p* = 1.28 × 10^−5^ ES = 0.42) and imcoh (*p* = 1.14 × 10^−5^, ES = 0.24). All the connectivity metrics allowed to observe a significant difference for the clustering coefficient, whilst five out of ten allowed to observe a significant difference for the assortativity and seven out of ten for the modularity.

The results from the WEI15 approach (where the 15% of weights were preserved) are summarized in Table 6. A significant difference between the two conditions in global efficiency was observed for seven out of the ten connectivity metrics. All the connectivity metrics allowed to observe a significant difference for the clustering coefficient, whilst four out of ten allowed to observe a significant difference for the assortativity and six out of ten for the modularity. In this case, four out of ten connectivity metrics, namely ciplv, pli, pli_unbiased and wpli_debiased, allowed to observe differences between the two conditions for all the network measures.

The results from the WEI50 approach (where the 50% of weights were preserved) are summarized in Table 7. A significant difference between the two conditions in global efficiency was observed for all the connectivity metrics. All the connectivity metrics allowed to observe a significant difference for the clustering coefficient, whilst five out of ten allowed to observe a significant difference for the assortativity and seven out of ten for the modularity. In this case, five out of ten connectivity metrics, namely ciplv, cohy, pli, pli_unbiased and wpli_debiased, allowed to observe differences between the two conditions for all the network measures.

The results from the ECO approach are summarized in Table 8. A significant difference between the two conditions in global efficiency was observed for the only one connectivity metric, namely cohy (*p* = 5.90 × 10^−4^, ES = 0.33). Three out of the ten connectivity metrics allowed to observe a significant difference for the clustering coefficient, whilst six out of ten allowed to observe a significant difference for the assortativity and none the modularity.

The results from the MST approach are summarized in Table 9. A significant difference between the two conditions in leaf fraction was observed for the six out of ten metrics. Six out of the ten connectivity metrics allowed to observe a significant difference for the kappa parameter, whilst none for the diameter, eccentricity and hierarchy parameters.

## 4. Discussion

In this study, we compared ten different connectivity metrics in a realistic scenario where two resting-state conditions, namely eyes-closed and eyes-open, were contrasted. As a first step, we performed a cluster analysis to understand how the metrics naturally arrange themselves into clusters. Later, to assess the possible differences induced by the different connectivity metrics, we reported the results in terms of statistical significance, effect size and direction of the effect (eyes-closed–eyes-open), using four different densities (preserving 10%, 15%, 50% and 100% of the weights) and two methods to filtering information in complex brain network that help to overcome the problem of network density in network analytical studies, namely the minimum spanning tree (MST) [20] and the efficiency cost optimization (ECO) [21].

The cluster analysis pointed out that the natural aggregation differs from the one we could initially expect (where each connectivity metric ideally represents a separate and distinct cluster). Indeed, if we consider the silhouette analysis results as the best way the different metrics naturally reorganize in clusters, we should conclude that it is possible to observe only two main clusters from the ten connectivity metrics. Nevertheless, as reported by the purity values across different k values, we have observed a strong variability in the quality of the clusters (expressed by the purity), suggesting that some connectivity metric spread over different clusters. In particular, if we take a closer look at the purity values for k equal to 10, it is possible to observe that some metrics, namely pli, coh, ppc and cohy, tend to be present in different clusters, while others, namely imcoh, plv and wpli, are mainly present into one single cluster. In any case, we do not think this latest finding may be considered as evidence of better quality of some specific metric over the others.

As for the network analysis, the main result of this study shows that different connectivity metrics, especially when thresholding approaches are implemented, may lead to relevant differences in the final outcomes, also in the case of a very simple realistic scenario where the underlying effect should be particularly straightforward [38,39,40]. In particular, the results show that for the clustering coefficient only it is possible to observe a statistical significance for all the connectivity metrics, but only in the case proportional thresholding is implemented. Moreover, the effect size shows a relevant variance among the different connectivity metrics and thresholding methods. A slightly more pronounced consistency among the connectivity metrics can be observed with a density increase, where for WEI15 and WEI50 the number of metrics that show similar results is higher. This is also confirmed by the results obtained using all the connections, thus preserving 100% of the weights, where we observed a significant difference over all the connectivity metrics. In contrast, the use of efficiency cost optimization and minimum spanning tree tend to amplify the differences between the connectivity metrics. It is, however, important to highlight that the direction of the effect is always consistent for all the metrics and for all the thresholding approaches.

In our opinion, these findings confirm that the (often arbitrary) choice of the adopted connectivity metric may have an important impact on the outcomes reported in the current literature on functional connectivity in EEG. As a consequence, we suggest caution when using the term functional connectivity interchangeably for different connectivity metric since this may lead to an erroneous belief of the generalizability of the results. We also would like to stress that this problem, the generalization of the results based on one arbitrary connectivity metric, may be also more relevant when the underlying effects are more subtle and less trivial (i.e., effects of treatment or comparison between healthy and pathological groups) or when the individual variability may have an even more robust effect [41,42].

An important limitation of the present study is related to the possible influence due to the source localization and parcellation methods. In fact, it has been previously shown, in a simulation study [14], that the choice of the inverse method and source imaging package may induce a considerable variability in the functional connectivity estimate. In any case, we may speculate that this possible effect adds even more variability and uncertainty on the reported findings. It is also important to highlight that there are several other issues that may play a relevant role in network analysis [43] that still remain to be addressed. Furthermore, it is even more important to stress the importance to replicate the reported findings in other EEG datasets to understand to what extent these results depend on this specific set of EEG recordings. Finally, it is also relevant to recognize that the use of thresholding approaches in functional networks are still debated since there are evidences that even weak connections are particularly meaningful. On the other hand, the interpretation of networks measures extracted from functional connectivity patterns (including the measures used in the present paper) are not easy to interpret and may be considered vague at least as the term functional connectivity that we debate in this study. This study was not intended to directly compare these metrics with the aim to understand if any of them outperforms the others or may represent the best choice to unveil specific network differences. In brief, we only would like to stress how the reported results, on any experimental design, may be affected by the arbitrary choice of the connectivity metric.

## 5. Conclusions

In conclusion, our results show that, even though all the metrics tend to show an effect on the same direction, source-level EEG functional connectivity estimates and the derived network measures may display a considerable dependency on the (often arbitrary) choice of the selected metric. This variability may reflect uncertainty and ambiguity in the final results, especially in less trivial scenarios. We suggest that this issue should be always discussed when reporting findings based on functional connectivity in EEG and ideally, it would be important to report and discuss the final outcomes based on more than one metric, making always explicit the adopted approach.

## Figures and Tables

**Figure 1 entropy-23-00005-f001:**
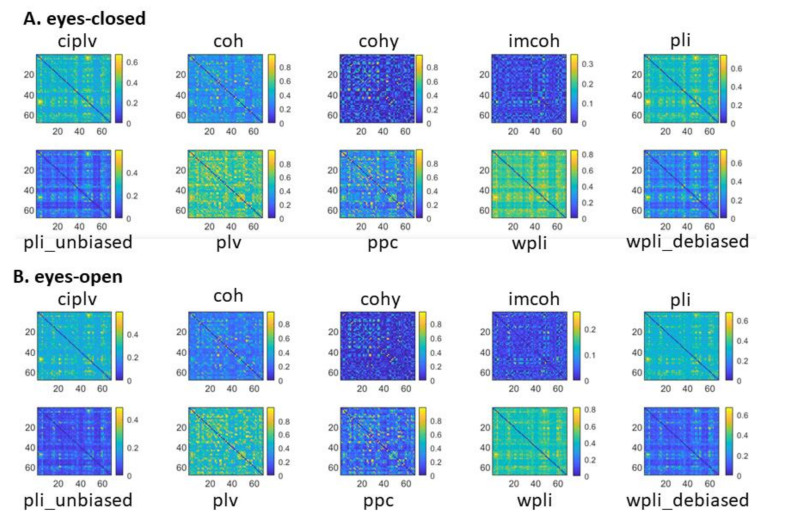
Global connectivity patterns (averaged over all the subjects) for each connectivity metric and for each condition: (**A**) eyes-closed; (**B**) eyes-open.

**Figure 2 entropy-23-00005-f002:**
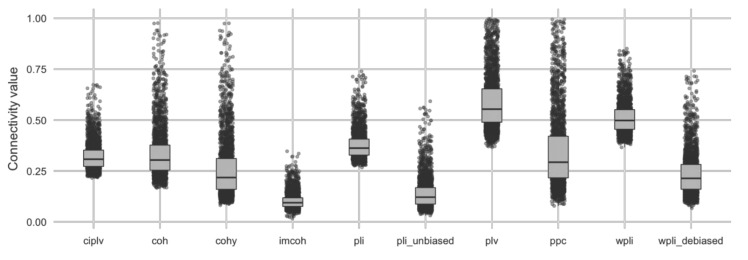
Weights distribution for all the connectivity metrics (weights are averaged across the 109 subjects).

**Figure 3 entropy-23-00005-f003:**
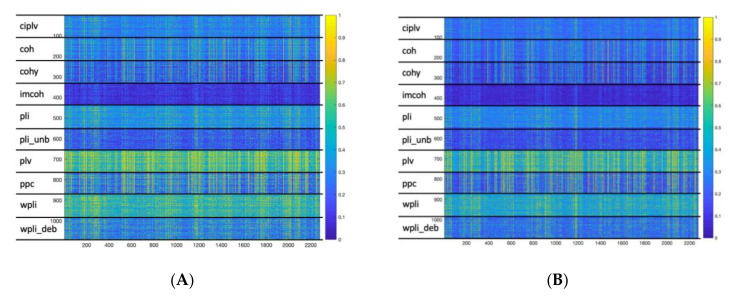
Connectivity profiles for all the connectivity metrics for eyes-closed (**A**) and eyes-open; (**B**) resting-state.

**Table 1 entropy-23-00005-t001:** Silhouette values for eyes-open and eyes-closed conditions at different k.

k	2	3	4	5	6	7	8	9	10
silhouette (eyes-closed)	0.413	0.314	0.343	0.305	0.271	0.232	0.221	0.191	0.177
silhouette (eyes-open)	0.455	0.290	0.349	0.318	0.300	0.261	0.210	0.186	0.164

**Table 2 entropy-23-00005-t002:** Purity values for eyes-open and eyes-closed conditions with k = 10.

Cluster	1	2	3	4	5	6	7	8	9	10
purity (eyes-closed)	0.361	0.885	0.364	0.939	0.391	0.590	0.657	0.869	0.733	0.982
purity (eyes-open)	0.905	0.980	0.427	1.000	0.780	0.517	0.593	0.459	0.915	0.983

**Table 3 entropy-23-00005-t003:** Purity computed using pseudo-labels. Eyes-closed (left columns) and eyes-open condition (right columns).

k.	Purity	Majority Cluster	Purity	Majority Cluster
2	0.854	[ciplv,coh,cohy,imcoh,pli2,ppc,wpli2]	0.999	[ciplv,coh,cohy,imcoh,pli,pli2,ppc,wpli2]
	0.756	[pli,plv,wpli]	0.709	[plv,wpli]
3	0.938	[ciplv,coh,cohy,pli,ppc,wpli2]	0.956	[ciplv,coh,cohy,pli,ppc,wpli2]
	0.788	[imcoh,pli2]	0.867	[imcoh,pli2]
	0.829	[plv,wpli]	0.945	[plv,wpli]
4	0.841	[ciplv,pli,wpli2]	0.856	[ciplv,pli,wpli2]
	0.823	[coh,cohy,ppc]	0.934	[coh,cohy,ppc]
	0.997	[plv,wpli]	1	[plv,wpli]
	0.900	[imcoh,pli2]	0.961	[imcoh,pli2]
5	0.627	[wpli]	0.750	[wpli]
	0.800	[plv]	0.932	[plv]
	0.837	[ciplv,pli,wpli2]	0.871	[ciplv,pli,wpli2]
	0.936	[imcoh,pli2]	0.994	[imcoh,pli2]
	0.997	[coh,cohy,ppc]	1	[coh,cohy,ppc]
6	0.649	[pli,wpli]	0.755	[wpli]
	0.716	[plv]	0.973	[plv]
	0.807	[ciplv,pli2,wpli2]	0.871	[ciplv,pli,wpli2]
	0.690	[imcoh]	0.994	[imcoh,pli2]
	0.549	[cohy]	0.562	[ppc]
	0.890	[coh,ppc]	0.814	[coh,cohy]
7	0.643	[pli,wpli]	0.748	[wpli]
	0.691	[plv]	0.973	[plv]
	0.806	[ciplv,pli2,wpli2]	0.871	[ciplv,pli,wpli2]
	0.690	[imcoh]	0.994	[imcoh,pli2]
	0.766	[coh,ppc]	0.897	
	0.744		0.876	[coh,ppc]
	0.626	[cohy]	0.611	[cohy]
8	0.658	[pli,wpli2]	0.815	[wpli]
	0.967	[plv]	0.982	[plv]
	0.724	[ciplv,pli2]	0.955	[ciplv,pli,wpli2]
	0.900	[imcoh]	0.991	[imcoh]
	0.759	[coh,ppc]	0.897	
	0.764		0.876	[coh,ppc]
	0.652	[cohy]	0.611	[cohy]
	0.864	[wpli]	0.593	[pli2]
9	0.475	[pli]	0.905	[wpli]
	0.967	[plv]	0.982	[plv]
	0.661	[ciplv,wpli2]	0.467	[pli]
	0.939	[imcoh]	1	[imcoh]
	0.759	[coh,ppc]	0.897	
	0.764		0.876	[coh,ppc]
	0.652	[cohy]	0.611	[cohy]
	0.869	[wpli]	0.740	[ciplv,wpli2]
	0.733	[pli2]	0.915	[pli2]
10	0.475	[pli]	0.905	[wpli]
	0.885		0.980	
	0.661	[ciplv,wpli2]	0.427	[pli]
	0.934	[imcoh]	1	[imcoh]
	0.391	[coh]	0.780	[ppc]
	0.560	[ppc]	0.517	[coh]
	0.657	[cohy]	0.593	[cohy]
	0.869	[wpli]	0.740	[ciplv,wpli2]
	0.733	[pli2]	0.915	[pli2]
	0.982	[plv]	0.983	[plv]

**Table 4 entropy-23-00005-t004:** Statistical results for the FWEI (100% of weights preserved) approach where eyes-open and eyes-closed conditions were contrasted. For each connectivity and network measure is reported the *p*-value, the effect size (ES) and the direction (D) of the effect.

FWEI	Global Efficiency			CC			Assortativity			Modularity		
	*p*-Values	ES	D	*p*-Values	ES	D	*p*-Values	ES	D	*p*-Values	ES	D
ciplv	1.94 × 10^−17^	0.81	EC < EO	5.38 × 10^−18^	0.83	EC > EO	ns			2.55 × 10^−4^	0.35	EC < EO
coh	1.11 × 10^−18^	0.85	EC < EO	6.82 × 10^−19^	0.85	EC > EO	ns			7.17 × 10^−16^	0.77	EC < EO
cohy	2.39 × 10^−17^	0.81	EC < EO	8.42 × 10^−18^	0.82	EC > EO	ns			1.58 × 10^−12^	0.68	EC < EO
imcoh	2.46 × 10^−17^	0.81	EC < EO	2.46 × 10^−17^	0.81	EC > EO	ns			9.26 × 10^−8^	0.51	EC < EO
pli	3.10 × 10^−17^	0.81	EC < EO	8.42 × 10^−18^	0.82	EC > EO	ns			9.20 × 10^−3^	0.25	EC < EO
pli_unbiased	1.69 × 10^−12^	0.68	EC < EO	7.85 × 10^−13^	0.69	EC > EO	ns			2.25 × 10^−7^	0.50	EC < EO
plv	2.73 × 10^−17^	0.81	EC < EO	7.38 × 10^−18^	0.82	EC > EO	ns			6.26 × 10^−15^	0.75	EC < EO
ppc	4.47 × 10^−16^	0.78	EC < EO	6.14 × 10^−18^	0.83	EC > EO	ns			3.23 × 10^−16^	0.78	EC < EO
wpli	2.80 × 10^−18^	0.84	EC < EO	1.95 × 10^−18^	0.84	EC > EO	ns			6.63 × 10^−7^	0.48	EC < EO
wpli_debiased	1.63 × 10^−16^	0.79	EC < EO	2.76 × 10^−15^	0.76	EC > EO	ns			5.15 × 10^−10^	0.60	EC < EO

**Table 5 entropy-23-00005-t005:** Statistical results for the WEI10 (10% of weights preserved) approach where eyes-open and eyes-closed conditions were contrasted. For each connectivity and network measure is reported the *p*-value, the effect size (ES) and the direction (D) of the effect.

WEI10	Global Efficiency			CC			Assortativity			Modularity		
	*p*-Values	ES	D	*p*-Values	ES	D	*p*-Values	ES	D	*p*-Values	ES	D
ciplv	ns			1.91 × 10^−7^	0.50	EC > EO	6.30 × 10^−5^	0.38	EC < EO	5.19 × 10^−3^	0.27	EC < EO
coh	ns			4.51 × 10^−11^	0.63	EC > EO	ns			ns		
cohy	1.28 × 10^−5^	0.42	EC < EO	2.06 × 10^−12^	0.67	EC > EO	ns			7.49 × 10^−3^	0.26	EC < EO
imcoh	1.14 × 10^−2^	0.24	EC < EO	8.05 × 10^−6^	0.43	EC > EO	ns			7.03 × 10^−3^	0.26	EC < EO
pli	ns			1.80 × 10^−6^	0.46	EC > EO	1.03 × 10^−4^	0.37	EC < EO	2.41 × 10^−3^	0.29	EC < EO
pli_unbiased	ns			7.79 × 10^−9^	0.55	EC > EO	1.14 × 10^−5^	0.42	EC < EO	3.05 × 10^−4^	0.35	EC < EO
plv	ns			5.69 × 10^−5^	0.39	EC > EO	ns			ns		
ppc	ns			1.25 × 10^−6^	0.46	EC > EO	ns			ns		
wpli	ns			1.24 × 10^−3^	0.31	EC > EO	1.07 × 10^−2^	0.24	EC < EO	1.61 × 10^−3^	0.30	EC < EO
wpli_debiased	ns			2.21 × 10^−7^	0.50	EC > EO	4.18 × 10^−3^	0.27	EC < EO	8.40 × 10^−4^	0.32	EC < EO

**Table 6 entropy-23-00005-t006:** Statistical results for the WEI15 (15% of weights preserved) approach where eyes-open and eyes-closed conditions were contrasted. For each connectivity and network measure is reported the *p*-value, the effect size (ES) and the direction (D) of the effect.

WEI15	Global Efficiency			CC			Assortativity			Modularity		
	*p*-Values	ES	D	*p*-Values	ES	D	*p*-Values	ES	D	*p*-Values	ES	D
ciplv	8.20 × 10^−3^	0.25	EC < EO	2.10 × 10^−8^	0.54	EC > EO	1.45 × 10^−5^	0.42	EC < EO	3.71 × 10^−4^	0.34	EC < EO
coh	3.16 × 10^−4^	0.35	EC < EO	2.39 × 10^−12^	0.67	EC > EO	ns			ns		
cohy	3.24 × 10^−10^	0.60	EC < EO	5.64 × 10^−13^	0.69	EC > EO	ns			1.26 × 10^−5^	0.42	EC < EO
imcoh	3.48 × 10^−4^	0.34	EC < EO	5.10 × 10^−11^	0.63	EC > EO	ns			ns		
pli	4.86 × 10^−3^	0.27	EC < EO	4.96 × 10^−9^	0.56	EC > EO	4.01 × 10^−5^	0.39	EC < EO	5.83 × 10^−5^	0.38	EC < EO
pli_unbiased	9.32 × 10^−5^	0.37	EC < EO	7.91 × 10^−12^	0.66	EC > EO	2.72 × 10^−6^	0.45	EC < EO	3.08 × 10^−5^	0.40	EC < EO
plv	ns			9.19 × 10^−7^	0.47	EC > EO	ns			ns		
ppc	ns			1.59 × 10^−8^	0.54	EC > EO	ns			ns		
wpli	ns			5.81 × 10^−6^	0.43	EC > EO	ns			9.26 × 10^−4^	0.32	EC < EO
wpli_debiased	8.88 × 10^−3^	0.25	EC < EO	2.49 × 10^−11^	0.64	EC > EO	9.96 × 10^−3^	0.25	EC < EO	1.17 × 10^−4^	0.37	EC < EO

**Table 7 entropy-23-00005-t007:** Statistical results for the WEI50 (50% of weights preserved) approach where eyes-open and eyes-closed conditions were contrasted. For each connectivity and network measure is reported the *p*-value, the effect size (ES) and the direction (D) of the effect.

WEI50	Global Efficiency			CC			Assortativity			Modularity		
	*p*-Values	ES	D	*p*-Values	ES	D	*p*-Values	ES	D	*p*-Values	ES	D
ciplv	3.10 × 10^−17^	0.80	EC < EO	<0.0001	0.78	EC > EO	ns			1.49 × 10^−6^	0.46	EC < EO
coh	1.30 × 10^−18^	0.84	EC < EO	<0.0001	0.79	EC > EO	1.84 × 10^−5^	0.41	EC < EO	4.25 × 10^−10^	0.59	EC < EO
cohy	2.46 × 10^−17^	0.81	EC < EO	<0.0001	0.75	EC > EO	7.39 × 10^−6^	0.42	EC < EO	2.89 × 10^−6^	0.44	EC < EO
imcoh	2.46 × 10^−17^	0.81	EC < EO	<0.0001	0.77	EC > EO	0.0008	0.31	EC < EO	2.97 × 10^−6^	0.44	EC < EO
pli	5.07 × 10^−17^	0.80	EC < EO	<0.0001	0.78	EC > EO	0.0014	0.30	EC < EO	0.0001	0.36	EC < EO
pli_unbiased	1.88 × 10^−12^	0.67	EC < EO	<0.0001	0.70	EC > EO	0.0004	0.33	EC < EO	3.15 × 10^−7^	0.48	EC < EO
plv	2.33 × 10^−16^	0.78	EC < EO	<0.0001	0.68	EC > EO	0.0002	0.34	EC < EO	ns		
ppc	5.19 × 10^−16^	0.77	EC < EO	<0.0001	0.72	EC > EO	3.60 × 10^−5^	0.39	EC < EO	2.67 × 10^−8^	0.53	EC < EO
wpli	5.82 × 10^−18^	0.82	EC < EO	<0.0001	0.78	EC > EO	0.0081	0.25	EC < EO	9.26 × 10^−6^	0.42	EC < EO
wpli_debiased	1.68 × 10^−16^	0.78	EC < EO	<0.0001	0.75	EC > EO	0.0070	0.25	EC < EO	2.91 × 10^−7^	0.49	EC < EO

**Table 8 entropy-23-00005-t008:** Statistical results for the ECO approach where eyes-open and eyes-closed conditions were contrasted. For each connectivity and network measure is reported the *p*-value, the effect size (ES) and the direction (D) of the effect.

ECO	Global Efficiency			CC			Assortativity			Modularity		
	*p*-Values	ES	D	*p*-Values	ES	D	*p*-Values	ES	D	*p*-Values	ES	D
ciplv	ns			ns			1.45 × 10^−4^	0.36	EC < EO	ns		
coh	ns			6.72 × 10^−3^	0.26	EC > EO	ns			ns		
cohy	5.91 × 10^−4^	0.33	EC < EO	ns			ns			ns		
Imcoh	ns			ns			3.87 × 10^−3^	0.28		ns		
pli	ns			ns			1.32 × 10^−4^	0.37	EC < EO	ns		
pli_unbiased	ns			ns			5.28 × 10^−4^	0.33	EC < EO	ns		
Plv	ns			1.56 × 10^−3^	0.30	EC > EO	ns			ns		
Ppc	ns			5.44 × 10^−3^	0.27	EC > EO	6.91 × 10^−3^	0.26		ns		
Wpli	ns			ns			9.61 × 10^−3^	0.25		ns		
wpli_debiased	ns			ns			ns			ns		

**Table 9 entropy-23-00005-t009:** Statistical results for the MST approach where eyes-open and eyes-closed conditions were contrasted. For each connectivity and network measure is reported the *p*-value, the effect size (ES) and the direction (D) of the effect.

MST	Leaf Fraction			Diameter			Eccentricity			Hierarchy			Kappa		
	*p*-Values	ES	D	*p*-Values	ES	D	*p*-Values	ES	D	*p*-Values	ES	D	*p*-Values	ES	D
ciplv	6.30 × 10^−4^	0.33	EC > EO	ns			ns			ns			5.91 × 10^−5^	0.38	EC > EO
coh	ns			ns			ns			ns			ns		
cohy	ns			ns			ns			ns			ns		
imcoh	4.00 × 10^−6^	0.44	EC > EO	ns			ns			ns			3.41 × 10^−8^	0.53	EC > EO
pli	7.25 × 10^−4^	0.32	EC > EO	ns			ns			ns			4.31 × 10^−7^	0.48	EC > EO
pli_unbiased	6.02 × 10^−4^	0.33	EC > EO	ns			ns			ns			6.79 × 10^−7^	0.48	EC > EO
plv	ns			ns			ns			ns			ns		
ppc	ns			ns			ns			ns			ns		
wpli	2.94 × 10^−5^	0.40	EC > EO	ns			ns			ns			4.88 × 10^−6^	0.44	EC > EO
wpli_debiased	1.92 × 10^−3^	0.30	EC > EO	ns			ns			ns			1.99 × 10^−5^	0.41	EC > EO

## Data Availability

All the EEG recordings are available at the following link: https://physionet.org/content/eegmmidb/1.0.0/.

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
