# Peer review of "On the Variability of Functional Connectivity and Network Measures in Source-Reconstructed EEG Time-Series"

_entropy, 2020, doi:10.3390/e23010005_

Round 1

Reviewer 1 Report

Authors in this manuscript demonstrate that the choice of the connectivity metric may have an important impact on the outcomes on functional connectivity in EEG. They suggest caution when using the term functional connectivity interchangeably for different connectivity characteristics.

The manuscript is well written; clear, and easy to understand. The text is slightly repetitive.

I am not an expert on some of the methods used (e.g. some EEG pre-processing techniques, cluster analysis), but it seems that a large amount of solid work was involved in the study.

The Discussion is a little harder to read than the rest of the text.

I think I get the general message of the article.

What I missed a bit was a simple closure for the specific application studied here. On the basis of all the tables and results, is it possible to say which connectivity measures seem to be the most promising and which the least (given the particular “eyes-closed – eyes-open” task)?

Reviewer 2 Report

The work tries to investigate a relevant problem in complex brain network studies: the reliability of the functional connectivity extraction techniques. Many authors already presented similar or wider works [e.g. https://pubmed.ncbi.nlm.nih.gov/23812847/], however, these authors aimed to quantitatively inspect the nature of the functional connectivity measure variability and this has a kind of originality. I have some major concerns that, however, can be addressed by a consistent revision of the manuscript:

  1. Authors made use of the term "correlation" many times and this is a source of confusion. Correlation is a precise mathematically defined function. What they, I assumed, want to refer to is the "statistical dependence" or "functional coherence" [as proposed by the seminal work of Pascal Fries 2005 "communication through coherence" that should be cited] that give the opportunity to consider under this term the complex measures which do not rely on linear correlation. Please, consider the use of the proposed alternatives.
  2. It is unclear why they chose to threshold with the levels 10, 15, 20 and 100. It should be much more coherent to follow either a linear or (I suggest) a logarithmic scheme (in the latter by replacing the 20 with the 50).
  3. It is unknown the a priori of the analyzed experimental study (open vs closed eyes): what we would expect from a functional connectivity perspective by contrasting eye-open vs closed? It is not clear in the work and this point is crucial to better understand how much the 10 measures differ from the truth and which is the best one.
  4. authors estimated the effect (and direction) size in a unusual way and they should use a post hoc analysis instead (Tukey test for instance)
  5. Figure 1 is not so informative because the scale bars are not uniform across the 10 adjacency matrices. Furthermore I suggest to produce also a set of box plots where on the x-axis there are the distributions of connectivity weights for each measure plus the average
  6. the stastical framework used to quantify differences can be improved with few efforts: I suggest to use a multivariate generalized linear model (i.e. y are the 64x64 edges) with 11 covariates (the 10 measures plus the contrast O/C eye). It will return a more rigorous analysis of what effectively edge changes and can answer to the question: are some connections between specific brain regions more subjected to variability in comparison to more stable ones?
  7. Paragraph at page 5 ends without an explanation of the results contained in table 3. Please, add it.
  8. Authors correctly proposed to use two accepted methods of edge filtering (ECO and MST) but they used them erroneously. Comparisons of networks without a filtering is meaningless. So I strongly suggest to repeat the analyses by filtering first the 100, 50, etc. and then perform the clustering analysis and network featuring. Eventually, to furtherly strengthen the effectiveness of edges, authors often produced surrogated data (e.g. by preserving the fourier transform of the original signal) and assess how many time out of 100 or 1000 the surrogated signals proposed that edge with a threshold of 5% (a sort of permutation "bootstrapped" test). It is a plus but they could also consider this technique.

Round 2

Reviewer 2 Report

Authors addressed all my concerns and the work is now ready to be published in my opinion.